# Navigating with Graph Representations for Fast and Scalable Decoding of Neural Language Models

**Minjia Zhang**    **Xiaodong Liu**    **Wenhan Wang**    **Jianfeng Gao**    **Yuxiong He**

Microsoft

`{minjiaz,xiaodl,wenhanw,jfgao,yuxhe}@microsoft.com`

## Abstract

Neural language models (NLMs) have recently gained a renewed interest by achieving state-of-the-art performance across many natural language processing (NLP) tasks. However, NLMs are very computationally demanding largely due to the computational cost of the decoding process, which consists of a softmax layer over a large vocabulary. We observe that in the decoding of many NLP tasks, only the probabilities of the top-$K$ hypotheses need to be calculated preciously and $K$ is often much smaller than the vocabulary size. This paper proposes a novel softmax layer approximation algorithm, called **F**ast **G**raph **D**ecoder (FGD), which quickly identifies, for a given context, a set of $K$ words that are most likely to occur according to a NLM. We demonstrate that FGD reduces the decoding time by an order of magnitude while attaining close to the full softmax baseline accuracy on neural machine translation and language modeling tasks. We also prove the theoretical guarantee on the softmax approximation quality.

## 1   Introduction

Drawing inspiration from biology and neurophysiology, recent progress on many natural language processing (NLP) tasks has been remarkable with deep neural network based approaches, including machine translation [1–3], sentence summarization [4], speech recognition [5–7], and conversational agents [8–11]. Such approaches often employ a neural language model (NLM) in a decoder at inference time to generate a sequence of tokens (e.g., words) given an input [1–7, 9, 10, 12–14].

One long-recognized issue of decoding using NLMs is the computational complexity, which easily becomes a bottleneck when the vocabulary size is large. Consider a beam search decoder using a NLM. At each decoding step, a recurrent neural network [15, 16] first generates a context vector based on each partial hypothesis in the beam. It then uses a softmax layer to compute a word probability distribution over the vocabulary [17–19]. The softmax layer consists of an inner product operator that projects the context vector into a vocabulary-sized vector of logits, followed by a softmax function that transforms these logits into a vector of probabilities. Finally, the decoder selects top-$K$ words with the highest probabilities given the context (i.e., *top-K maximum subset* of inner product), and stores the expended hypotheses and their probabilities in the beam. The most computationally expensive part in this process is the softmax layer, where the complexity of performing inner product is linear with respect to the vocabulary size. In this paper we strive to develop new softmax approximation methods for fast decoding.

Many techniques have been proposed to speed up the softmax layer in training, such as hierarchical softmax [20] and sampling-based approaches [21–24]. However, most of them cannot be directly applied to decoding at inference because they rely on knowing the words to be predicted and they still need to calculate the probability of all words to find the most likely prediction. Other works speed up softmax inference (in training and decoding) by reducing the cost of computing each word's probability using some approximation [23, 25–27]. Though the cost of computing each word's probability is reduced, the complexity of softmax layer as a whole is still linear with respect to the size of the vocabulary. We notice that in many NLP tasks we only need to identify the top-$K$ most

likely next words given a context. Do we have to go over the entire large vocabulary to search for the top-$K$ words? Our answer is no. Before we present our approach, we review briefly the finding in biological science that motivates our research.

In spite of the large number of words in a vocabulary, the human brain is capable of managing them effectively and navigating the massive mental lexicon very efficiently. How is it possible? How is the vocabulary implemented in human brain? One of the theories from biological science indicates that human language has a character of complex network [28–30], where the intrinsic relational structure, which refers to the fact that words are related to each other and thus form a *small world graph*, provides some hints how the lexicon is mentally organized. To predict the next word given a context, humans never need to examine every word in a vocabulary stored in their brains. Instead, a person can immediately identify a small set of $K$ candidate words that are most semantically related to the context, and then picks the most proper word among the top-$K$ candidates. We believe that if we can represent the vocabulary using a similar *small world graph*, we can significantly improve the decoding efficiency of NLMs because softmax only needs to explicitly compute the probabilities of $K$ words, where $K$ is much smaller than the vocabulary size.

We propose a **F**ast **G**raph **D**ecoder (FGD) to approximate the softmax layer of a NLM in the decoding process. First, we construct a *small world graph* representation [31, 32] of a vocabulary. The nodes in the graph are words, each being represented using a continuous vector transformed from its word embedding vector in the NLM. The edges in the graph encode the word-word distances in a well-defined metric space. Then, at each decoding step, we identify for a given context (e.g., a partial hypothesis in the beam search) the top-$K$ hypotheses and compute their probabilities in the softmax layer of the NLM. We prove that finding the top-$K$ hypotheses in the softmax layer is equivalent to finding the $K$ nearest neighbors using FGD in the small world graph, and the latter can be performed approximately using efficient graph navigating methods. We also prove that the decoding error due to the use of the approximated $K$ nearest neighbor search with graph navigation is theoretically bounded.

We validate the effectiveness of our approach on two NLP tasks, neural machine translation and language modeling. Empirical results show that FGD achieves an order of magnitude speedup while attaining the accuracy in comparison with existing state-of-the-art approaches.

In the rest of the paper, Section 2 details the softmax layer implementation and the challenge. Section 3 describes FGD and gives theoretical justifications. Section 4 presents experimental results. Conclusions are drawn in Section 5.

## 2    Motivation and Challenge

The softmax layer of a NLM results in a major computational bottleneck at the decoding process in many NLP tasks. Consider a NLM that uses a two–layer LSTM and a vocabulary size of $|V|$ [17–19]. The total number of floating point operations (FLOPS) per LSTM step is 2(layer) $\times (I + D) \times D \times 4 \times 2$ [1], where $I$ and $D$ represent the input and hidden dimension, respectively. The number of FLOPS of the softmax layer is $D \times |V| \times 2$, which is proportional to the vocabulary size $|V|$. Assuming that the input/hidden dimension of the LSTM is 500 and the vocabulary size is 50K, the LSTM part has 8M FLOPS, whereas the softmax layer has 50M FLOPS. The softmax layer dominates the computational cost of the NLM, and even more so with a larger vocabulary.

This decoding bottleneck limits NLMs' application in many interactive services such as Web search, online recommendation systems and conversational bots, where low latency, often at the scale of milliseconds, is demanded. In addition, unlike model training where we can leverage the massive parallelism power of GPUs, inference needs to run in various clients ranging from PC, mobile, to IoT (Internet of Things) device, most of which have limited hardware resources and where GPUs are not always available [33]. Therefore, fast decoding is crucial to broaden the applicability of NLMs.

## 3    Approach

The goal of this work is to reduce the computational complexity of decoding, the major bottleneck for many NLP tasks. Our approach is called FGD (**F**ast **G**raph **D**ecoder), which consists of an offline

step that transforms the pre-trained word embeddings to a small world graph representation and an online inference step that finds the top-$K$ hypotheses, as outlined in Figure 1. In what follows, we will present in turn

- Why do we use the small world graph representation to find top-$K$ hypotheses? (Section 3.1)

- How to construct a small world graph? (Section 3.2)

- How to identify top-$K$ hypotheses for a given context on small world graph? (Section 3.3)

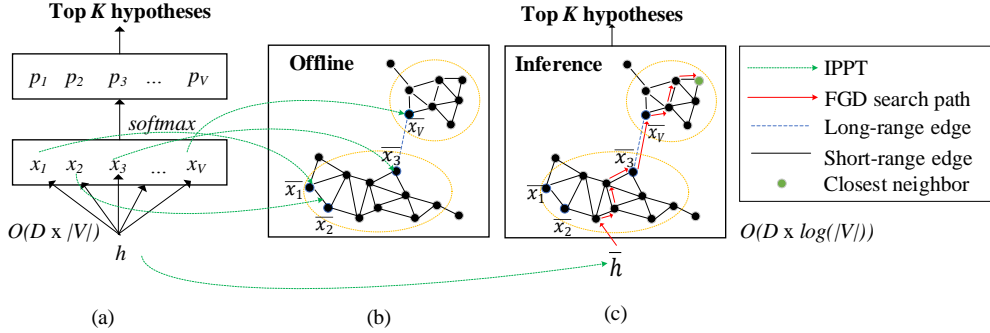

Figure 1: **Overview of FGD:** (a) illustrates the original operations of selecting top-$K$ hypotheses in the decoding process, where the top-$K$ hypotheses are selected from the output of $softmax$. The inner product between the context vector $h$ and word embedding vectors $x_1, x_2, .., x_{|V|}$ in the softmax layer is the most expensive part, which has a complexity of $O(D \times |V|)$. (b) shows the transformation from $x_1, x_2, .., x_{|V|}$ to a small world graph representation that encodes pair-wise similarity information among words. This transformation, incurring once offline, is essential for FGD to perform fast decoding. (c) shows the decoding process of FGD at online inference time. For a given context vector $h$, FGD identifies top-$K$ hypotheses in the small world graph and produces their probabilities with a search complexity of $O(D \times \log |V|)$.

### 3.1 Why Small World Graphs?

Nearest neighbor search is a commonly used method to identify top-$K$ points in a set that is most similar to a given point. The small world graph has been recently introduced to address the problem of nearest neighbor search [34, 35]. Research shows that navigation in small world graph exhibits $O(\log N)$ search complexity ($N$ represents the number of nodes in the graph), and performs well in high dimensionality [34–36]. These results motivate us to investigate the use of the small world graph to develop fast decoding methods for NLM with large vocabularies.

To get logarithmic search complexity, the small world graph needs to perform neighborhood selection based on well-defined pair-wise *metric* distance, as well as small world properties, such as great local connectivity (as in a lattice graph) combined with a small graph diameter (as in a random graph) [37], which we will describe in details in the following sections.

### 3.2 Small World Graph Construction

Figure 1 illustrates the transformation from the set of word embedding vectors in Figure 1(a) to a small world graph representation $G$ in Figure 1 (b). Given a set of word embedding vectors $X = [x_1, x_2, ..., x_{|V|}], x_i \in \mathbb{R}^D$, where $|V|$ represents the vocabulary size and $D$ is the word embedding dimension, our transformation process takes two steps.

1. **Inner product preserving transformation** (representation transformation): For each vector–based embedding $x_i$ in $X$, we apply a transformation called Inner Product Preserving Transformation (IPPT) to obtain $\overline{X} = [\overline{x_1}, \overline{x_2}, ..., \overline{x_{|V|}}], \overline{x_i} \in \mathbb{R}^{D+2}$ that establishes the equivalence of finding top-$K$ maximum subset of inner product in $X$ and searching for top-$K$ nearest neighbors in $\overline{X}$ with a given distance metric $\rho$. (Section 3.2.1)

2. **Small world graph construction** (data structure transformation): Denote $G = (\overline{X}, E)$ as the small world graph with the set $\overline{X}$ as graph nodes and $E$ as the graph edges, where we impose the set $E$ based on the distance metric $\rho$ to form a small world graph. (Section 3.2.2)

### 3.2.1 Inner Product Preserving Transformation

**Why is inner product insufficient?** Using inner product to represent the mutual similarity between nodes is deficient because it lacks very basic properties that are required to hold for distance (i.e., the inverse of similarity) functions in metric spaces (e.g., Euclidean spaces) – identity of indiscernibles and triangle inequality [38]. For example, under the Euclidean space, two points are the same iff their distance is 0. The inner product of a point $x$ to itself is $\|x\|^2$, but there can be other points whose inner product to $x$ is smaller than $\|x\|^2$. The search process on small world graphs relies on these properties to converge and achieve their efficiency [34].

**Representation transformation.** To create similarity relationship between words represented with a metric, we present a new method called Inner Product Preserving Transformation (IPPT) to convert the word embedding vectors to higher dimensional vectors. We establish the equivalence of finding top-$K$ maximum subset of inner product and searching for top-$K$ nearest neighbor with a distance metric in the higher dimension space. We use the notation $\langle \cdot, \cdot \rangle$ for the inner product and $\rho(\cdot, \cdot)$ for the distance in Euclidean space. Thus $\rho(\xi, \eta) = \|\xi - \eta\|_2 = \sqrt{\langle \xi - \eta, \xi - \eta \rangle}$. In the following lemma, we define a transform $\phi$ for the word embeddings and another transform $h \mapsto \overline{h}$ for the context vector into a higher dimensional space, such that the inner product between the embedding and context vectors remains the same before and after the transform. We call it *inner product preserving*. We include the proof in Appendix A.

**Lemma 3.1.** *Suppose $D \geq 1$. Let $x_i \in \mathbb{R}^D$ and $b_i \in \mathbb{R}$ be the vector-based word embedding and bias at position $i$ in the softmax layer respectively, for $1 \leq i \leq |V|$. Choose $U$ such that $U \geq \max_{i \in V} \sqrt{\|x_i\|_2^2 + b_i^2}$. Let $[;]$ represents vector concatenation. Define $\phi : \{(\xi, \eta) \in \mathbb{R}^D \times \mathbb{R} : \|\xi\|_2^2 + \eta^2 \leq U^2\} \longrightarrow \mathbb{R}^{D+2}$ as $\phi(x, b) = \left[x; b; \sqrt{U^2 - \|x\|_2^2 - b^2}\right]$. For a context vector $h \in \mathbb{R}^D$, let $\overline{h} = [h; 1; 0] \in \mathbb{R}^{D+2}$. Then for any $i \in V$, we have $\langle h, x_i \rangle + b_i = \langle \overline{h}, \phi(x_i, b_i) \rangle = \frac{1}{2}\left(U^2 + 1 + \|h\|_2^2 - \rho(\overline{h}, \phi(x_i, b_i))^2\right)$.*

The above lemma states that for a fixed context vector $h$, $\langle h, x_i \rangle + b_i$ depends linearly and monotonically decreasingly on $\rho(\overline{h}, \phi(x_i, b_i))^2$. In particular, $\langle h, x_i \rangle + b_i \leq \langle h, x_j \rangle + b_j$ iff $\rho(\overline{h}, \phi(x_i, b_i)) \geq \rho(\overline{h}, \phi(x_j, b_j))$. This gives rise to the equivalence between the top-$K$ maximum inner product search and top-$K$ nearest neighbor search in graph.

**Definition 1** (**Top-$K$ maximum (minimum) subset**). *Let $V$ be the set of vocabulary, and $1 \leq K \leq |V|$. We call $\mathcal{K}$ a top-$K$ maximum (minimum) subset for a function $f : V \to \mathbb{R}$, if $|\mathcal{K}| = K$ and $f(v_i) \geq f(v_j)$ $(f(v_i) \leq f(v_j))$ for all $v_i \in \mathcal{K}$ and $v_j \notin \mathcal{K}$.*

**Theorem 3.2.** *Suppose $1 \leq K \leq |V|$ and consider a fixed context vector $h$. Let $\mathcal{K} \subseteq V$ be a top-$K$ maximum subset for $v_i \mapsto \langle h, x_i \rangle + b_i$. Then $\mathcal{K}$ is also a top-$K$ minimum subset for the Euclidean distance $v_i \mapsto \rho(\overline{h}, \phi(x_i, b_i))$.*

Based on the theorem, we can build a small world graph in $\mathbb{R}^{D+2}$ to equivalently solve the top-$K$ maximum subset of inner product search problem.

### 3.2.2 Small World Graph Construction

We present the algorithm FGD–P (P for P̲reprocessing) for constructing the small world graph in Algorithm 1. FGD–P is performed only once on a trained model. It transforms the trained vector-based word representation $X$, which is a $D$-by-$|V|$ matrix, into $\overline{X}$ with IPPT (line 4–9).

FGD–P builds an in-memory proximity graph possessing small world properties using $G = CreateSwg(\overline{X}, M)$ (line 10). Several existing work have devoted to constructing the small world graph. Among the most accomplished algorithms, HNSW (Hierarchical Navigable Small Worlds) has recently attained outstanding speed-accuracy trade-offs [35]. We employ HNSW's graph construction algorithm to create a small world graph. We briefly describe the main ideas below and please find more details in Malkov and Yashunin [35].

| **Algorithm 1** | Offline preprocessing algorithm FGD–P |
| --- | --- |

1: **Input:** Trained weights of the softmax layer $X$, and bias vector b.
2: **Output:** Small world graph G, and $U_{max}$.
3: **Hyperparameter:** Small world graph neighbor degree M.
4: **for all** $i$ **in** $(0..|X| - 1)$ **do**
5:      $\tilde{X}_{:i} \leftarrow [X_{:i}; b_i]$             $\triangleright$ Word embedding and bias fusion
6: $U_{max} \leftarrow \max_i \|\tilde{X}_{:i}\|_2$
7: **for all** $i$ **in** $0..(|\tilde{W}| - 1)$ **do**
8:      $\Delta_i \leftarrow \sqrt{U_{max}^2 - \|\tilde{X}_{:i}\|_2^2}$             $\triangleright$ Calculate the normalizer
9:      $\overline{X}_{:i} \leftarrow [\tilde{X}_{:i}; \Delta_i]$
10: $G \leftarrow CreateSwg(\overline{X}, M)$             $\triangleright$ Build small world graph

The $CreateSwg$ creates a multi-layer small world graph $G$, which consists of a chain of subsets $V = L_0 \supseteq L_1 \supseteq \ldots \supseteq L_l$ of nodes as "layers" and the ground layer $L_0$ contains the entire $\overline{x_i}$ as nodes. $L_0$ is built incrementally by iteratively inserting each word vector $\overline{x_i}$ in $\overline{X}$ as a node. Each node will generate $M$ (i.e., the neighbor degree) out-going edges. Among those, $M - 1$ are *short-range* edges, which connect $\overline{x_i}$ to $M - 1$ sufficiently close nodes, i.e., neighbors, according to their pair-wise Euclidean distance to $\overline{x_i}$ (e.g., the edge between $\overline{x_1}$ and $\overline{x_2}$ in Figure 1 (b)). The rest is a *long-range* edge that connects $\overline{x_i}$ to a randomly picked node, which does not necessarily connect two closest nodes but connects isolated clusters (e.g., the edge between $\overline{x_3}$ and $\overline{x_{|V|}}$ in Figure 1 (b)). It is theoretically justified that these two types of edges give the graph small world properties [34, 35, 37]. Given $L_0$, each layer of $L_k(k > 0)$ is formed recursively by picking each node in $L_{k-1}$ with a fixed probability $1/M$, and the top layer contains only a single node. The number of layers is bounded by $O\left(\log |V| / \log M\right)$ [34].

### 3.3 Decoding as Searching Small World Graphs

FGD–I (Algorithm 2) shows how FGD is used to enable fast decoding (as in Figure 1 (c)). It first transforms the context vector $h$ to $[h; 1; 0] \in \mathbb{R}^{d+2}$(line 4). $SearchSwg(G, h, K)$ searches the small world graph to find the top-$K$ hypotheses using the search method from HNSW [35]. Here we provide a brief description of the methodology.

The search of the graph starts from its top layer $L_l$ and uses greedy search to find the node with the closest distance to $\overline{h}$ as an entry point to descend to the lower layer. The upper layers route $\overline{h}$ to an entry point in the ground layer $L_0$ that is already close to the nearest neighbors to $\overline{h}$. Once reaching the ground layer, $SearchSwg$ employs a prioritized breath-first search: It examines its neighbors and stores all the visited nodes in a priority queue based on their distances to the context vector. The length of the queue is bounded by $efSearch$, a hyperparameter that controls the trade-off of search time and accuracy. When the search reaches a stop condition (e.g., above a given number of distance calculation), $SearchSwg$ returns the results of top-$K$ word hypotheses and their distances to $\overline{h}$. We transform the distance value back to their inner product (line 5– 7) with the inner product preserving property of IPPT. FGD–I generates the output by computing a softmax distribution over the inner product of the top-$K$ returned results (line 8).

| **Algorithm 2** | Online inference algorithm FGD–I |
| --- | --- |

1: **Input:** Context vector $h$, small world graph $G$, and $U_{max}$.
2: **Output:** Probability distribution $P$ over top-$K$ word hypotheses.
3: **Hyperparameter:** Candidate queue length $efSearch$.
4: $\overline{h} \leftarrow [h; 1; 0]$             $\triangleright$ Map context vector from $\mathbb{R}^D$ to $\mathbb{R}^{D+2}$
5: $I^K, D^K \leftarrow SearchSwg(G, \overline{h}, K)$      $\triangleright$ Return top-$K$ hypotheses with minimal distance to $\overline{h}$
6: **for all** $i$ **in** $0..(K - 1)$ **do**
7:      $S[I_{:i}^K] \leftarrow \frac{1}{2}\left(\|\overline{h}\|_2^2 + U_{max}^2 - D_{:i}^{K^2}\right)$      $\triangleright$ Map Euclidean distance back to inner product
8: $P \leftarrow exp(S)/\sum exp(S)$             $\triangleright$ Compute top-$K$ softmax probability distribution

**Bounding softmax probability distribution.** In practice, letting $K = |V|$ is both slow and unnecessary, an approximated approach is often much more efficient without sacrificing much accuracy. We demonstrate empirically the effectiveness of our approach in Section 4 and provide a theoretically derived error bound of softmax approximation with top-$K$ word hypotheses toward a probability distribution in Appendix C.

## 4 Evaluation

**Summary of main results.** In this section, we present the results of FGD on two different tasks: neural machine translation (NMT) and language modeling (LM).

1. On NMT, FGD obtains more than 14X speedup on softmax layer execution time over full-softmax with a similar BLEU score to the baseline, and obtains 30X speedup at the cost of decreasing 0.67 BLEU score.

2. On LM, FGD scales with a logarithmic increase of execution time and outperforms full-softmax in speed by an order of magnitude for large vocabularies.

**Setup.** We implement FGD in Python using numpy[2]. To construct the small world graph, we employ a state-of-the-art framework NMSLIB [35, 39]. The execution time is given as the averaged per-step decoding time in milliseconds, measured on a 64-bit Linux Ubuntu 16.04 server with two Intel Xeon CPU E5-2650 v4 @ 2.20GHz processor with single thread regime so that all algorithms are compared under the same amount of hardware resource.

### 4.1 Neural Machine Translation

NMT is implemented using a sequence–to–sequence model which contains an RNN encoder and an RNN decoder. The decoder contains an output projection at every step to predict the next word. Decoding time and BLEU score [40] are the two major metrics for this evaluation. The lower the decoding time without sacrificing much BLEU, the better the result is. We trained a global attention-based [41] encoder–decoder model with two stacked unidirectional LSTM [1, 2] using the OpenNMT-py toolkit [42] on the IWSLT'14 German-English corpus [43]. We set the LSTM hidden dimension size to 200. The model is optimized with SGD using an initial learning rate of 1.0 and a dropout [44] ratio of 0.3. The dataset is tokenized and preprocessed using the OpenNMT data preprocessor with $|V| = 50,000$ frequent words [24, 41]. BLEU score is computed using the Moses toolkit [45].

Once the model is trained, we processed the trained weights in the softmax layer using FGD–P offline. It takes three minutes on our server to construct the small world graph. During online processing, the hyperparameter, $efSearch$, decides the length of the candidate queue to track nearest neighbors, which offers the trade-off between the online decoding speed and the BLEU score quality. We tested different $efSearch$ values and identified [20, 200] as a good range.

**Decoding time and BLEU score comparison with existing methods.** Two approaches are used for comparison: 1) a full-softmax approach; 2) a state-of-the-art approach, called SVD-softmax [25]. SVD-softmax improves the inference speed by approximating softmax layer using singular vector decomposition (SVD). It includes two steps: it first estimates the probability of each word using a small part of the softmax layer weight matrix, and then performs a refinement on top-$\overline{V}$ most likely words based on the previous estimated results. It reduces the complexity from $O(|V| \times D)$ to $O(|V| \times \overline{D} + |\overline{V}| \times D)$, where $1 \leq \overline{D} < D$. As suggested by [25], we use two configurations of SVD-softmax: *SVD-a*[3] and *SVD-b*[4].

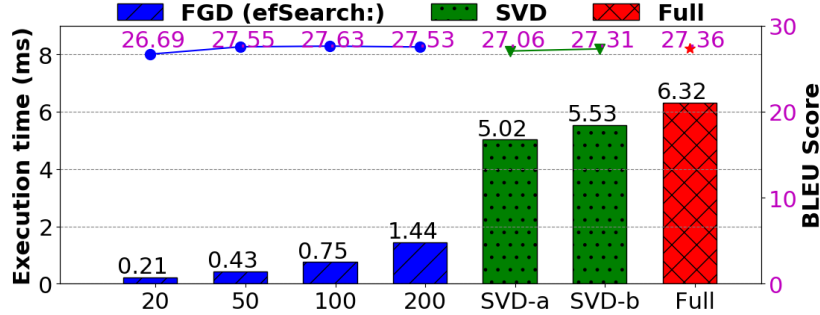

Figure 2: Execution time of the softmax layer and BLEU score of NMT model with FGD, SVD-softmax (**SVD**), and Full-softmax (**Full**). [20, 50, 100, 200] are the hyperparameter of $efSearch$ in FGD. Execution time is displayed as the height of the bar chart, in millisecond (lower is better). BLEU scores are labeled with colored numbers on the top (higher is better).

Figure 2 shows the main results — FGD achieves significantly lower execution time than the existing methods with comparable BLEU scores.

Comparing with full softmax, when $efSearch$ is 20, FGD reduces the execution time from 6.3ms to 0.21ms, achieving 30X speedup at the cost of losing 0.67 BLEU score. By increasing $efSearch$ to 50, FGD obtains nearly the same BLEU score as the full-softmax baseline, while reducing the execution time from 6.3ms to 0.43ms, achieving more than 14X speedup.

For SVD-softmax, we also observed that *SVD-b* approaches a BLEU score close to the full-softmax baseline, but it is much slower than FGD in terms of the execution time (5.53ms vs 0.43ms). *SVD-a* shows slightly better performance than *SVD-b* but with a lower BLEU score. Although the theoretical speedup of *SVD-a* is 5.5X, it gets only 1.3X speedup in practice because top-$\overline{V}$ most likely words selected in the first step appear at discontinuous location on memory, which causes non-negligible memory copy cost to bring them to a continuous space for the second step calculation.

**Sensitivity of sequence lengths.** Figure 3 reports the results with $efSearch = 100$. FGD is on a par with the full softmax baseline uniformly on different lengths (without statistically significant difference). It demonstrates the robustness of the proposed approach.

**Sensitivity of beam sizes.** We vary the beam size among 1, 2, 5, 10, which are typical settings used by prior work [1–3, 46]. Table 1 shows that, when $efSearch$ is equal or larger than 50, FGD obtains the BLEU scores close to the full softmax baseline under all beam sizes without and statistically significant difference.

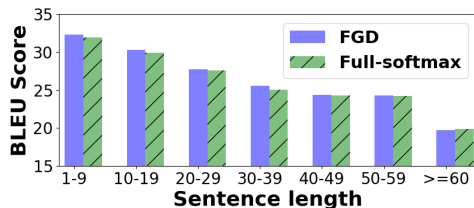

Figure 3: BLEU score breakdown by sentence length (setting $efSearch$=100).

| efSearch | Beam = 1 | Beam = 2 | Beam = 5 | Beam = 10 |
|---|---|---|---|---|
| 20 | 26.69 | 27.65 | 27.81 | 27.62 |
| 50 | 27.55 | 28.76 | 29.06 | 28.9 |
| 100 | 27.63 | 28.94 | 29.28 | 29.1 |
| 200 | 27.53 | 28.99 | 29.28 | 29.22 |
| Full | 27.36 | 28.91 | 29.45 | 29.34 |

Table 1: BLEU score on NMT task, with various beam sizes.

**Internals of FGD.** To reveal the internals of FGD, we analyzed two metrics, *precision@K* (or equivalently *P@K*) and *dist_cnt*. *Precision@K* measures the proportion of overlap between retrieved top-$K$ hypotheses and expected top-$K$ hypotheses, based on what top-$K$ on a full-softmax would return. *dist_cnt* measures the number of distance computation in FGD under a given $efSearch$. Table 2 reports *precision@K* when $K$ is 1, 2, 5, and 10, which correspond to beam size 1, 2, 5, and 10 respectively, and *dist_cnt* with versus without FGD. Overall, FGD achieves fairly high precision. In particular, gradually increasing $efSearch$ leads to higher precision at the expense of increased number of distance computation. This matches the observation that higher $efSearch$ leads to higher

BLEU score (Figure 2) and also longer execution time (Table 1). Further increasing $efSearch$ leads to little extra precision improvement but significantly more distance computation because the precision is getting close to 1, which explains why FGD can get close to baseline BLEU score (Table 1). We also observe that under the same $efSearch$, further increasing $K$ sometimes leads to slightly worse precision if $efSearch$ is not big enough (e.g., $efSearch$ is 20), as the highest ranked words not visited during the graph search are definitely lost. On the other hand, the computation of distance grows proportional to the increase of $efSearch$. Comparing with the full-softmax, the amount of distance computation is reduced by 10–50 times, which explains the speedup of decoding time (Figure 2).

| efSearch | P@1 | P@2 | P@5 | P@10 | dist_cnt (FGD/ Full) |
|---|---|---|---|---|---|
| 20 | 0.939 | 0.934 | 0.929 | 0.918 | 981 / 50K |
| 50 | 0.974 | 0.974 | 0.973 | 0.971 | 1922 / 50K |
| 100 | 0.986 | 0.986 | 0.987 | 0.987 | 3310 / 50K |
| 200 | 0.992 | 0.993 | 0.994 | 0.994 | 5785 / 50K |

Table 2: Precision and distance computation results on the NMT model.

## 4.2 Language Modeling

This section evaluates the impact of vocabulary sizes and word embedding dimensions on FGD using language models [5] trained on WikiText-2 [47]. The model uses a two–layer LSTM[6].

**Impact of vocabulary size.** We explore multiple models with different vocabulary size of 10,000 (10K), 20,000 (20K), 40,000 (40K), and 80,000 (80K). The vocabulary is created by tokenizing raw texts via Moses toolkit [45] and choosing the correspondingly topmost frequent words on the raw WikiText-2 dataset [47]. Both input and hidden dimension are set to 256.

Table 3 shows the impact on search quality by varying the vocabulary size from 10K to 80K. With the same $efSearch$, FGD generally obtains better precision results for smaller vocabularies; With the same vocabulary size, bigger $efSearch$ is better for high precision. With $efSearch$ being 200, the precision of FGD is getting very close to 100%.

| |V| | P@K | FGD (efSearch) | | | |
|---|---|---|---|---|---|
| | | 20 | 50 | 100 | 200 |
| 10K | P@1 | 0.870 | 0.938 | 0.989 | 1.000 |
| | P@10 | 0.909 | 0.972 | 0.992 | 0.998 |
| 20K | P@1 | 0.845 | 0.932 | 0.975 | 0.995 |
| | P@10 | 0.871 | 0.955 | 0.987 | 0.997 |
| 40K | P@1 | 0.808 | 0.912 | 0.936 | 0.980 |
| | P@10 | 0.845 | 0.931 | 0.961 | 0.991 |
| 80K | P@1 | 0.832 | 0.933 | 0.966 | 0.982 |
| | P@10 | 0.858 | 0.945 | 0.978 | 0.994 |

Table 3: Precision of FGD on WikiText-2 dataset varying vocabulary size.

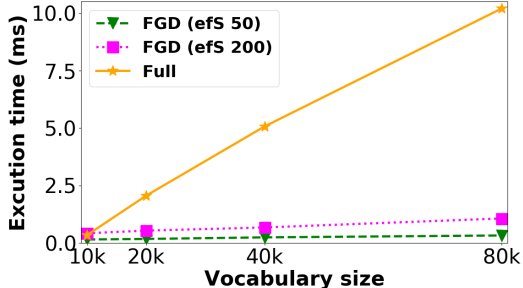

Figure 4: Scalability of WikiText-2 language model varying vocabulary size.

Figure 4 shows the decoding time of varying vocabulary sizes on the full softmax baseline and FGD (settings $efSearch$={50, 200} for the sake of readability). As expected, the execution time all increases with the increase of the vocabulary size. However, compared to the baseline, FGD provides a shorter execution time consistently. As the vocabulary size increases, the execution time of the baseline increases almost linearly, whereas FGD's execution time increases much more slowly. This is because the complexity of softmax is $O(D \times |V|)$, which is linear to the size of the vocabulary, whereas the complexity of FGD is $O(D \times \log |V|)$ is logarithmic to $|V|$. Therefore, FGD scales much

better and the improvement becomes more significant with larger vocabulary sizes. In particular, FGD is more than an order of magnitude faster than the baseline when the vocabulary size is medium or large. For example, FGD achieves more than 30X speedup with $|V| = 80K$ when $efSearch = 50$ (Appendix 6 includes a speedup graph).

It is worth mentioning that perplexity is often used to evaluate language models but it is not applicable here. This work focuses on generating the probabilities of the top-$K$ word hypotheses for fast decoding, and FGD optimizes that by saving tons of probability computation for the words that are not in the top-$K$ list. As a result, the evaluation in perplexity is not applicable because the probabilities of the words in the test set are undefined if they are not in the top-$K$ list. We observe that many end applications such as machine translation do not require the probabilities of the words that are not in the top-$K$ list. That is why we evaluate the LM using precision@K and the accuracy on the end applications.

**Sensitive of word embedding dimension.** We also tested various word embedding dimensions, where FGD gets higher precision consistently with an order of magnitude execution time reduction in comparison with baselines (see Appendix D).

## 5   Discussion

**Compatibility with parallelizable recurrence.** There has been work on optimizing language models and its end applications by leveraging parallelizable structures to speed up recurrent layer, because the sequential dependency in the recurrent layer is hard to be computed in parallel [50–56]. In comparison with these work, FGD speeds up the vocabulary searching process at the softmax layer. By incorporating FGD with these approaches, together they will reduce the end-to-end decoding time even further.

**Generalization to training.** There are several challenges that need to be addressed to make training efficient using FGD. During inference, weights of the softmax layer are static. In contrast, those weights are constantly changing as new examples are seen during training. It would be prohibitively slow to update parameters for all word vector embeddings and update the underline small world graph after every training step. One possibility is, during backpropagation, to propagate gradients based on top-$K$ hypotheses that are retrieved during the forward pass of the model and update the parameter vectors of only these retrieved hypotheses. Then the challenge is to figure out how gradients based on only a small set of examples affect word embedding vectors and how these sparse gradients can be propagated in a computationally efficient way.

## 6   Conclusion

We propose a novel decoding algorithm, called **F**ast **G**raph **D**ecoder (FGD), which quickly navigates, for a given context, on a *small world graph* representation of word embeddings to search for a set of $K$ words that are most likely to be the next words to predict according to NLMs. On neural machine translation and neural language modeling tasks, we demonstrate that FGD reduces the decoding time by an order of magnitude (e.g., 14X speedup comparing with the full softmax baseline) while attaining similar accuracy. As the further work, we also like to explore how to speed up NLMs training with large vocabularies.

## Acknowledgments

We thank Kevin Duh for reading a previous version of this paper and providing feedback. We thank the anonymous reviewers for their helpful suggestions for improving this paper.

## Footnotes

[1]It times 4 because an LSTM has 3 gates and 1 memory cell, and it times 2 because each weight value causes a multiply–and–add operation.

[2]http://www.numpy.org/

[3]The preview window width $\overline{D}$ is set to 16, and the refinement window width $\overline{V}$ is set to 2500.

[4]The preview window width $\overline{D}$ is set to 16, and the refinement window width $\overline{V}$ is set to 5000.

[5] https://github.com/pytorch/examples/tree/master/word_language_model

[6] The models are trained with stochastic gradient descent (SGD) with an initial learning rate of 20 [48]. The batch size is set to 20, and the network is unrolled for 35 timesteps. Dropout is applied to LSTM layers with a dropout ratio of 0.3 [44]. Gradient clipping is set to 0.25 [49].

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
