[Supplementary Material]

## A    Proof of Lemma and Theorem

*Lemma 3.1 proof.* For the first equation can be proved by expanding the inner product according to the definition of the transformation. Note that $\langle \tilde{h}, \phi(x_i, b_i) \rangle = \left\langle [h; 1; 0], \left[ x_i; b_i; \sqrt{U^2 - \|x_i\|^2 - b_i^2} \right] \right\rangle = \langle h, x_i \rangle + 1 \cdot b_i + 0 \cdot \sqrt{U^2 - \|x_i\|^2 - b_i^2} = \langle h, x_i \rangle + b_i$. The second equation follows from the relation between the inner product and the distance in Euclidean spaces: $\rho(\tilde{h}, \phi(x_i, b_i))^2 = \|\tilde{h} - \phi(x_i, b_i)\|^2 = \|\tilde{h}\|^2 + \|\phi(x_i, b_i)\|^2 - 2\langle \tilde{h}, \phi(x_i, b_i) \rangle = \|h\|^2 + 1 + U^2 - 2\langle \tilde{h}, \phi(x_i, b_i) \rangle$. □

*Theorem 3.2 proof.* By Lemma 3.1, the distance $\rho(\tilde{h}, \phi(x_i, b_i))$ depends strictly monotonically decreasingly on $\langle h, x_i \rangle + b_i$. The claim of the theorem then follows straightforwardly upon this fact. □

## B    Visualization of Similarity Relation

Our IPPT (Section 3.2.1) approach provides a convenient way to transform inner product against word embedding vectors to a metric space, exploiting closeness between words through their metric relation, and then inversely transforming metric distance back to inner product result in $O(1)$ complexity, which then can be consumed by the final softmax function to compute word probabilities. To see the metric relation, we utilize t-SNE [57] to visualize the weights at the vocabulary projection layer by mapping it to a 2D Euclidean distance space. As shown in Figure 5, words in the projection layer clearly has conceptual and positional properties. Thus semantic more similar words are closer in distance.

Figure 5: Projection of word embedding vectors on a 2D Euclidean distance space. 400 words are sampled from the vocabulary projection weights of WikiText-2 by filtering the high frequency words (higher than 1000) and low frequency words (less than 100). Training methodology to get the weights is included in Section 4.2.

## C    A Bound on Top-$K$ Softmax Approximation

In this section we derive an error bound of top-$K$ highest ranked words approximation on softmax toward a probability distribution. Let $V$ be the set of vocabulary. Let $s_i$ ($1 \leq i \leq |V|$) be the scores produced using exhaustive search, sorted in decreasing order. Suppose $L$ is a lower bound for $\{s_i\}$, i.e., $L \leq \min s_i$. Typically one chooses $L$ to be a negative number with sufficiently large absolution value. If there is no known lower bound, one may also set $L = -\infty$ and agrees that $\exp(-\infty) = 0$. The probability distribution generated by applying softmax on $s_i$ is given by $p_i = \exp(s_i) / \sum_i \exp(s_i)$. Using exhaustive search we are able to compute $p_i$ as exact. However

with approximated techniques, we are only able to obtain an approximation $\hat{p}_i$ of the distribution. In real applications we typically only care about how $\hat{p}_i$ differs from $p_i$ for the top-$K$ words. The error in such approximations comes from two sources: (1) the accuracy of the approximation to obtain the approximated top-$K$; and (2) the approximation of $\sum_i \exp(s_i)$. In the following theorem we give a quantitative analysis of how large the relative error could be.

**Theorem C.1.** *Let $V$, $s_i$, $L$, $p_i$ be as above. Let $\mathcal{K} \subseteq \{1, \ldots, |V|\}$ be the ground truth top-$K$ indices. Suppose an approximation top-$K$ softmax algorithm gives $\mathcal{K}' \subseteq \{1, \ldots, |V|\}$ as the approximated top-$K$ indices. Let $\hat{s}_i$ be the approximated score the algorithm assigns to the $i$-th word in $V$, and let $s' = \min_{i \in \mathcal{K}'} \hat{s}_i$. Assume that (i) the algorithm assigns exact scores $\hat{s}_i = s_i$ to those $i \in \mathcal{K}'$; and (ii) it assigns a score $\hat{s}_i$ for $i \notin \mathcal{K}'$ such that $L \leq \hat{s}_i \leq s'$. The approximated probability distribution is given by $\hat{p}_i := \exp(\hat{s}_i)/\sum_i \exp(\hat{s}_i)$. Let $\mathcal{K}'' = \{i : s_i \geq s'\}$. Then the relative error of probability distribution approximation is bounded by $|\hat{p}_i - p_i|/p_i \leq \dfrac{\sum_{i \in \mathcal{K}'' \setminus \mathcal{K}'} (\exp(s_i) - \exp(L)) + (|V| - |\mathcal{K}''|)(\exp(s') - \exp(L))}{\sum_{i \in \mathcal{K}'} \exp(\hat{s}_i) + (|V| - K) \exp(L)}$ for any $i \in \mathcal{K} \cap \mathcal{K}'$.*

*Proof.* First note that $p_i = \dfrac{\exp(s_i)}{\sum_i \exp(s_i)}$ and that $\hat{p}_i = \dfrac{\exp(\hat{s}_i)}{\sum_i \exp(\hat{s}_i)}$. Since $i \in \mathcal{K} \cap \mathcal{K}'$, we have $s_i = \hat{s}_i$. We then deduce that

$$\frac{|\hat{p}_i - p_i|}{p_i} = \frac{|\sum_i \exp(s_i) - \sum_i \exp(\hat{s}_i)|}{\sum_i \exp(\hat{s}_i)}. \tag{1}$$

We then proceed to bound both the numerator and the denominator.

To find an upper bound for the numerator, first note that

$$\left| \sum_i \exp(s_i) - \sum_i \exp(\hat{s}_i) \right| \leq \left| \sum_{i \in \mathcal{K}''} \exp(s_i) - \sum_{i \in \mathcal{K}''} \exp(\hat{s}_i) \right| + \left| \sum_{i \notin \mathcal{K}''} \exp(s_i) - \sum_{i \notin \mathcal{K}''} \exp(\hat{s}_i) \right|. \tag{2}$$

For the first summand, first observe that $\mathcal{K} \subseteq \mathcal{K}''$ and $\mathcal{K}' \subseteq \mathcal{K}''$. Therefore

$$\left| \sum_{i \in \mathcal{K}''} \exp(s_i) - \sum_{i \in \mathcal{K}''} \exp(\hat{s}_i) \right| = \left| \sum_{i \in \mathcal{K}'} (\exp(s_i) - \exp(\hat{s}_i)) + \sum_{i \in \mathcal{K}'' \setminus \mathcal{K}'} (\exp(s_i) - \exp(\hat{s}_i)) \right|.$$

The condition (i) implies that the first sum is zero. The second sum is always non-negative since $\exp(s_i) \geq \exp(s') \geq \exp(\hat{s}_i)$ for $i \in \mathcal{K}'' \setminus \mathcal{K}'$. Thus $\left| \sum_{i \in \mathcal{K}'' \setminus \mathcal{K}'} (\exp(s_i) - \exp(\hat{s}_i)) \right| = \sum_{i \in \mathcal{K}'' \setminus \mathcal{K}'} (\exp(s_i) - \exp(\hat{s}_i)) \leq \sum_{i \in \mathcal{K}'' \setminus \mathcal{K}'} (\exp(s_i) - \exp(L))$. For the second summand in the right hand side of Equation (2), note that $\left| \sum_{i \notin \mathcal{K}''} \exp(s_i) - \sum_{i \notin \mathcal{K}''} \exp(\hat{s}_i) \right| \leq \sum_{i \notin \mathcal{K}''} |\exp(s_i) - \exp(\hat{s}_i)|$. Since both $s_i, \hat{s}_i \in [L, s']$ for all $i \notin \mathcal{K}''$, we have $\sum_{i \notin \mathcal{K}''} |\exp(s_i) - \exp(\hat{s}_i)| \leq \sum_{i \notin \mathcal{K}''} (\exp(s') - \exp(L)) = (|V| - |\mathcal{K}''|)(\exp(s') - \exp(L))$. This shows the upper bound of the numerator.

For the denominator in the right hand side of Equation (1), simply note that $\sum_i \exp(\hat{s}_i) = \sum_{i \in \mathcal{K}'} \exp(\hat{s}_i) + \sum_{i \notin \mathcal{K}'} \exp(\hat{s}_i) \geq \sum_{i \in \mathcal{K}'} \exp(\hat{s}_i) + (|V| - K) \exp(L)$. This then concludes the proof of the theorem. $\qquad\square$

It is worthy to point out that the numerator of the above error bound can be rewritten as $\sum_{i \in \mathcal{K}'' \setminus \mathcal{K}'} (\exp(s_i) - \exp(s')) + (|V| - K)(\exp(s') - \exp(L))$. Intuitively, the theorem states that the accuracy of the softmax probability approximation for the top-$K$ words depends on three quantities: (i) $\sum_{i \in \mathcal{K}'' \setminus \mathcal{K}'} (\exp(s_i) - \exp(s'))$, which measures how many words are "missed" by the approximation of top-$K$ words. (ii) $\exp(s') - \exp(L)$, which measures the distribution of the scores found by the approximation. The smaller (i) and (ii) are (relative to $\sum_{i \in \mathcal{K}'} \exp(\hat{s}_i)$), the better the approximation is.

We also observe that when the precision at $K$ is 1 for the approximation algorithm, then the bound depends only on the sum of exponential scores and the smallest top-$K$ score retrieved by the algorithm.

**Corollary C.1.1.** *Let the notations be the same as in the above theorem. Assume that the precision at $K$ of the approximation is 1. Further assume that all the scores $s_i$ are distinct. Then the relative error to the approximated softmax probability distribution is bounded above by $|\hat{p}_i - p_i|/p_i \leq \frac{(|V| - K)(\exp(s') - \exp(L))}{\sum_{i \in \mathcal{K}'} \exp(\hat{s}_i) + (|V| - K)\exp(L)}$ for any $i \in \mathcal{K} \cap \mathcal{K}'$.*

*Proof.* It suffices to show that $\mathcal{K}'' = \mathcal{K}'$. Since the precision at $K$ is 1, we have $\mathcal{K} = \mathcal{K}'$, which means $s' = \min_{i \in \mathcal{K}'} \hat{s}_i = \min_{i \in \mathcal{K}} s_i$. Now assume $i \in \mathcal{K}''$, then by definition of $\mathcal{K}''$, $s_i \geq \min_{j \in \mathcal{K}} s_j$. Since all scores are distinct, this shows that $s_i$ is amongst the top-$K$, i.e., $s_i \in \mathcal{K} = \mathcal{K}'$. Thus $\mathcal{K}'' \subseteq \mathcal{K}'$. The other direction of inclusion is trivial. □

# D   Additional Results

**Speedup with different vocabulary size.**   Figure 6 shows the speedup for FGD (with different $efSearch$) over the execution time of full-softmax for vocabulary size 10K, 20K, 40K, and 80K. When $efSearch = 20$, FGD achieves more than 65X speedup over the baseline with a vocabulary size 80K. Even with smaller vocabulary size, FGD still achieves roughly an order of magnitude speedup. Overall, FGD achieves speedup over the baseline consistently and scales well with different $efSearch$ values.

Figure 6: Performance of FGD, normalized to full-softmax execution time. Higher is better.

**Sensitivity of word embedding dimension.**   Table 4 reports the precision with varying word vector embedding dimension 128, 256, and 512 on the WikiText-2 language modeling. The vocabulary size is set to the default 33,728. In most cases, $efSearch$ being 50 or 100 is sufficient to provide high precision (e.g., $> 0.95$). Over 0.99 precision can be reached when $efSearch$ is 200. This indicates that FGD offers high precision with different word embedding dimensions.

| D | P@K | FGD (efSearch) | | | |
|---|-----|------|------|------|------|
|   |     | 20   | 50   | 100  | 200  |
| 128 | P@1  | 0.913 | 0.993 | 0.998 | 0.999 |
|     | P@10 | 0.819 | 0.934 | 0.976 | 0.992 |
| 256 | P@1  | 0.832 | 0.917 | 0.958 | 0.992 |
|     | P@10 | 0.866 | 0.944 | 0.976 | 0.995 |
| 512 | P@1  | 0.854 | 0.921 | 0.968 | 0.988 |
|     | P@10 | 0.884 | 0.950 | 0.979 | 0.995 |

Table 4: Precision of FGD on WikiText-2 dataset varying word vector embedding dimension.

Figure 7 compares with the execution time of FGD and full-softmax, FGD achieves an order of magnitude reduction of execution time.

Figure 7: Scalability of WikiText-2 language model varying word vector embedding dimension.