[Reviews · NeurIPS 2018]

Reviewer 1



Thanks for the feedback. To clarify, parallel NMT in the worked I pointed to get rids of auto-regressivity and not just recurrence. I agree it's orthogonal but could be the fastest NMT system --------- The paper develops a method for fast computation of top-k hypotheses (words in this paper) during decoding for natural language generation. In an offline pre-processing step, a graph is constructed with words as nodes in the graph and edges using parameters of a model trained in the usual way. The paper provides thorough experiments comparing to previous methods in the literature. The speedup offered by these methods is impressive. Other Comments: 1) There has been lot of recent work on parallel neural machine translation (https://arxiv.org/abs/1711.02281 ; https://arxiv.org/abs/1805.11063). Importantly, beam search is not required by these methods but computing the softmax is still necessary. It would be interesting to see how the proposed speedups compare to them and see if both these directions can be combined. 2) It would be useful to include decode time numbers in Table 1 3) Even though attention methods generally do not require k-best lists, there are some methods that do top-k attention instead of full attention (https://arxiv.org/abs/1607.01426). Having experiments on these models would make the paper even stronger.

Reviewer 2



Update after author response: Thanks for the detailed response! It's a strong submission and I vote for an accept. =========== This paper aims to speed up the computation of the softmax over a large vocabulary, which is quite common in some NLP tasks like e.g., language modeling. Specifically, the proposed method formulates the problem into a nearest neighbor search in a small world graph, and applies a log time algorithm to find the approximate top K predictions. The resulting time complexity reduces to logarithmic in the vocabulary size in expectation, in contrast to the linear one in a standard softmax. The proposed method is empirically compared to both full softmax and a state-of-the-art baseline on language modeling and neural machine translation. Experimental results show that the method achieves significant speed-up, without sacrificing too much accuracy, and that it scales to larger-size models. The paper is overall well-motivated and clearly-written, and I find it interesting to read. My major concern is perhaps how the method generalizes to training, detailed below. Concerns: - Small world graph construction: It would be nice to see some discussions on the complexity of the graph construction step, and whether it could become a potential problem if the algorithm is applied to training. - Probabilities: Are there any good reasons to not report perplexities in the language modeling task? It appears to me that in a FGD, it is less straightforward to assign probabilities to the corpus. One thing on top of my head is to attribute most of the mass to the retrieved top K, with some additional tricks to avoid numerical issues. Have the authors tried this? - The title is a bit confusing, since language models'' usually mean the language modeling task, but here the algorithm applies to many other tasks involving a large softmax. - There are several statements that look like finding top K in sublinear time complexity,'' especially around lines 30--40. I find such statements a bit misleading, since essentially one would need approximations to achieve this.

Reviewer 3



This paper proposes to use small world graphs as data structures to embed words vectors into for a fast lookup of potentially relevant words that are then put into a softmax. It's a useful datastructure. Not quite machine learning... but useful for softmax classifications in MT and language modeling. Wish it was trained on a fast and competitive language model and actually mention perplexity numbers. Ideally you can use: https://github.com/salesforce/awd-lstm-lm On NMT, it would have been better to use a parallelizable transformer network also. In spite of the large number of words in a vocabulary, human brain is capable --> In spite of the large number of words in a vocabulary, THE human brain is capable speedup while attaining the accuracy --> speedup while maintaining the accuracy